# Temporal Trends of Major Bleeding and Its Prediction by the Academic Research Consortium-High Bleeding Risk Criteria in Acute Myocardial Infarction

**DOI:** 10.3390/jcm11040988

**Published:** 2022-02-14

**Authors:** Sungwook Byun, Eun Ho Choo, Gyu-Chul Oh, Sungmin Lim, Ik Jun Choi, Kwan Yong Lee, Su Nam Lee, Byung-Hee Hwang, Chan Joon Kim, Mahn-Won Park, Chul Soo Park, Hee-Yeol Kim, Ki-Dong Yoo, Doo Soo Jeon, Ho Joong Youn, Wook Sung Chung, Min Chul Kim, Myung Ho Jeong, Hyeon-Woo Yim, Youngkeun Ahn, Kiyuk Chang

**Affiliations:** 1Division of Cardiology, Department of Internal Medicine, Bucheon St. Mary’s Hospital, College of Medicine, The Catholic University of Korea, Seoul 06591, Korea; bsw625@daum.net (S.B.); cumckhy@catholic.ac.kr (H.-Y.K.); 2Division of Cardiology, Department of Internal Medicine, Seoul St. Mary’s Hospital, College of Medicine, The Catholic University of Korea, Seoul 06591, Korea; cmcchu@catholic.ac.kr (E.H.C.); david.gyuchul@gmail.com (G.-C.O.); cycle210@catholic.ac.kr (K.Y.L.); hbhmac@naver.com (B.-H.H.); younhj@catholic.ac.kr (H.J.Y.); chungws@catholic.ac.kr (W.S.C.); 3Division of Cardiology, Department of Internal Medicine, Uijeongbu St. Mary’s Hospital, College of Medicine, The Catholic University of Korea, Seoul 06591, Korea; mdsungminlim@gmail.com (S.L.); godandsci@catholic.ac.kr (C.J.K.); 4Division of Cardiology, Department of Internal Medicine, Incheon St. Mary’s Hospital, College of Medicine, The Catholic University of Korea, Seoul 06591, Korea; mrfasthand@catholic.ac.kr (I.J.C.); coronary@catholic.ac.kr (D.S.J.); 5Division of Cardiology, Department of Internal Medicine, St. Vincent’s Hospital, College of Medicine, The Catholic University of Korea, Seoul 06591, Korea; yellow-night@hanmail.net (S.N.L.); yookd@catholic.ac.kr (K.-D.Y.); 6Division of Cardiology, Department of Internal Medicine, Daejeon St. Mary’s Hospital, College of Medicine, The Catholic University of Korea, Seoul 06591, Korea; pmw6193@catholic.ac.kr; 7Division of Cardiology, Department of Internal Medicine, Yeouido St. Mary’s Hospital, College of Medicine, The Catholic University of Korea, Seoul 06591, Korea; charlie@catholic.ac.kr; 8Division of Cardiology, Department of Internal Medicine, Chonnam National University Hospital, Chonnam National University School of Medicine, Gwangju 61469, Korea; kmc3242@hanmail.net (M.C.K.); myungho@chollian.net (M.H.J.); cecilyk@hanmail.net (Y.A.); 9Clinical Research Coordinating Center, Department of Preventive Medicine, College of Medicine, The Catholic University of Korea, Seoul 06591, Korea; y1693@catholic.ac.kr

**Keywords:** acute myocardial infarction, percutaneous coronary intervention, bleeding risk

## Abstract

Limited data exist on the temporal trend of major bleeding and its prediction by the Academic Research Consortium-High Bleeding Risk (ARC-HBR) criteria in acute myocardial infarction (AMI) patients undergoing percutaneous coronary intervention (PCI). We investigated 10-year trends of major bleeding and predictive ability of the ARC-HBR criteria in AMI patients. In a multicenter registry of 10,291 AMI patients undergoing PCI between 2004 and 2014 the incidence of Bleeding Academic Research Consortium (BARC) 3 and 5 bleeding was assessed, and, outcomes in ARC-defined HBR patients with AMI were compared with those in non-HBR. The primary outcome was BARC 3 and 5 bleeding at 1 year. Secondary outcomes included all-cause mortality and composite of cardiovascular death, myocardial infarction, or ischemic stroke. The annual incidence of BARC 3 and 5 bleeding in the AMI population has increased over the years (1.8% to 5.8%; *p* < 0.001). At 1 year, ARC-defined HBR (*n* = 3371, 32.8%) had significantly higher incidence of BARC 3 and 5 bleeding (9.8% vs. 2.9%; *p* < 0.001), all-cause mortality (22.8% vs. 4.3%; *p* < 0.001) and composite of ischemic events (22.6% vs. 5.8%; *p* < 0.001) compared to non-HBR. During the past decade, the incidence of major bleeding in the AMI population has increased. The ARC-HBR criteria provided reliable predictions for major bleeding, mortality, and ischemic events in AMI patients.

## 1. Introduction

Bleeding in patients with acute myocardial infarction (AMI) is not uncommon and is a major concern as much as ischemic events after percutaneous coronary intervention (PCI) [1,2,3,4]. In the contemporary PCI era, when treatment with drug eluting stent (DES) and more potent P2Y_12_ inhibitor is implemented, the epidemiology of major bleeding in the AMI population is expected to be far different from the past. However, data on the temporal trend of major bleeding in this population are scarce. In addition, the risk of bleeding in AMI patients is higher than in those with chronic coronary syndrome or unstable angina [5,6,7]. Therefore, predicting bleeding risk and implementing appropriate antithrombotic strategy in AMI patients are more important than in those with other coronary artery diseases. Currently, however, no bleeding risk stratification system has been established to guide antithrombotic therapy in AMI. 

Recently, the Academic Research Consortium (ARC) developed consensus-based criteria to identify high bleeding risk (HBR) patients after PCI [8]. According to the ARC-HBR criteria, patients who meet at least one major criterion or two minor criteria are classified as HBR. To date, four large-scale cohort studies have validated the ARC-HBR criteria in all-comers PCI registries [9,10,11,12]. However, the study population in those studies consisted mainly of patients with chronic coronary syndrome or unstable angina, and thus included relatively small proportions of AMI patients. As a result, the predictive ability of the ARC-HBR criteria is little known in AMI, which is distinct from angina in terms of its clinical and pathophysiological aspects. Hence, this study sought to investigate 10-year trends of major bleeding and epidemiologic change in related risk factors, as well as the predictive ability of the ARC-HBR criteria in real-world, large-scale AMI populations.

## 2. Materials and Methods

### 2.1. Study Population

The CardiOvascular Risk and idEntificAtion of potential high-risk population in Acute Myocardial Infarction (COREA-AMI) registry was designed to evaluate the long-term clinical outcomes of consecutive AMI patients from January 2004 to August 2014 who were treated with PCI at nine major university cardiac centers throughout South Korea. The definition of AMI and the inclusion and exclusion criteria in the COREA-AMI registry and are described in Appendix A. Factors recorded in this observational registry include demographic, clinical, and procedural data, as well as short- and long-term clinical outcomes until 2019. Informed consent was obtained from all of the participating patients. The study protocol conformed with the Declaration of Helsinki regarding investigations in humans and was approved by the Institutional Review Board of each participating centre. This registry has been registered on www.ClinicalTrials.gov as NCT02806102.

### 2.2. Application of the ARC-HBR Criteria 

Among the original ARC-HBR criteria, definitions of some criteria were modified, and a few criteria such as chronic bleeding diathesis, planned major surgery, and recent major surgery or trauma within 30 days before PCI were not applied to the study due to lack of data. The adopted definitions of each criterion in this study are shown in Appendix A. The patients were classified as the HBR or non-HBR group according to the criteria [8]. The HBR group consisted of patients who met at least one major or two minor criteria. The remaining patients were considered non-HBR. In addition, since the ARC-HBR criteria are essentially bleeding risk factors, their prevalence over time in the AMI population was also investigated. 

### 2.3. Treatment and Data Collection 

Coronary angiography and interventions were performed according to standard guidelines [13,14,15,16]. Patients naïve to aspirin and P2Y_12_ inhibitors were administered loading doses of aspirin and P2Y_12_ inhibitors including clopidogrel, prasugrel, or ticagrelor. Revascularization techniques, devices, and adjunctive antithrombotic therapy were at the discretion of the operators. Treatments after PCI included maintenance doses of aspirin and P2Y_12_ inhibitors. Optimal pharmacological therapy, including statins, beta-blockers, or renin-angiotensin system inhibitors, was recommended according to the guidelines. Doses were titrated, and medications were changed during follow-up, if needed due to each patient’s condition. 

Data were prospectively collected on a web-based system after eliminating personal information. Independent research personnel collected data, and independent reviewers and cardiologists assessed angiographic and procedural data. Clinical events and outcomes were obtained from electronic medical records and telephone conversations. All adverse clinical events were confirmed by the clinical event adjudication committee of the cardiovascular (CV) centre of Seoul St. Mary’s Hospital, Republic of Korea. Especially, all bleeding events before 2012 were retrospectively re-adjudicated according to the Bleeding Academic Research Consortium (BARC) classification, since the BARC scale was introduced in 2011 [17]. Mortality was verified by the National Health Insurance Service, the government-managed insurance program in Korea that includes the entire population of the country. 

### 2.4. Clinical Outcomes 

Primary outcome was cumulative incidence of BARC 3 and 5 bleeding 1 year after PCI. The primary outcome was also used to evaluate the temporal trend of major bleeding incidence during the study period and the predictive ability of the ARC-HBR criteria, compared with Predicting Bleeding Complications in Patients Undergoing Stent Implantation and Subsequent Dual Antiplatelet Therapy (PRECISE-DAPT) score [18]. Bleeding events within 1 month of index PCI were considered periprocedural bleeding. Secondary outcomes included cumulative incidence of all-cause mortality at 1 year, as well as a composite of CV death or MI or ischemic stroke. CV death was defined as death resulting from AMI, sudden cardiac death, heart failure, stroke, or other vascular cause. Ischemic stroke was defined as an episode of neurologic dysfunction related to the brain, spinal cord, or retinal vascular injury as a result of infarction.

### 2.5. Statistical Analysis 

Categorical variables were expressed as frequencies (percentages) and were compared using chi-square test. This test was also used to evaluate the difference in the prevalence of bleeding risk factors over time. Continuous variables were expressed as mean ± standard deviation and compared by Student’s *t*-test. To assess unadjusted temporal trend of major bleeding, the Mann-Kendall trend test was used. Cumulative incidences of primary and secondary outcomes were compared between HBR and non-HBR patients using log-rank test. The Cox regression proportional hazard model analyzed the impact of HBR on incidence rate. The same statistical methods were used in subgroups analysis such as non-ST segment elevation myocardial infarction (NSTEMI), ST segment elevation myocardial infarction (STEMI), and patients treated with aspirin and potent P2Y_12_ inhibitor. The discriminating ability of the ARC-HBR criteria was evaluated using C-statistics and compared with PRECISE-DAPT score by Delong test. The additive prognostic effect of the major ARC-HBR criteria was analyzed by classifying patients into four groups according to the number of fulfilled criteria: non-HBR, ≥2 minor without major, 1 major with any minor, and ≥2 major with any minor. Statistical analyses were conducted with R version 4.0.2 (R Foundation for Statistical Computing, Vienna, Austria). All statistical testing was two-sided. A *p*-value < 0.05 was considered statistically significant.

## 3. Results

### 3.1. Study Population

Of the 10,719 AMI patients who underwent PCI from January 2004 to August 2014, 10,291 (96.0%) patients completed 1-year follow-up and were analyzed in this study. 428 patients were lost to follow-up within 1 year after index procedure. All data on the ARC-HBR criteria applied in this study and outcomes were identifiable.

### 3.2. Baseline Characteristics 

Of the 10,219 patients with AMI who underwent PCI, 3371 (32.8%) were classified as HBR by the ARC-HBR definition. Compared with non-HBR, patients in the HBR group were older, were more likely to be female, and had a lower body mass index (Table 1). In addition, the prevalence of comorbidities such as diabetes, hypertension, and peripheral artery disease were higher in the HBR group. Heart disease, such as previous MI, coronary artery bypass grafting, and atrial fibrillation were more prevalent in the HBR group. In terms of clinical presentation, STEMI was less common in the HBR group, and patients in the HBR group were more likely to have higher Killip class. Regarding procedure, femoral access, the use of bare-metal stent, complex PCI, and PCI of the left main coronary artery were more frequent in the HBR group. However, the number of treated lesions, total stent number and length did not differ significantly between groups. At the time of discharge, the rate of dual antiplatelet therapy (DAPT) and potent P2Y_12_ inhibitor use were lower in the HBR group, whereas proton pump inhibitors were prescribed more frequently to HBR patients.

### 3.3. Trends in Incidence of Major Bleeding in AMI Population and Types of PCI

Figure 1A and Appendix A show the unadjusted temporal trends in the incidence of BARC 3 and 5 bleeding within 1 year after PCI in AMI population between 2004 and 2014, and the composition of types of PCI in each year. Over the years, the annual BARC 3 and 5 bleeding incidence has increased from 1.8% to 5.8% (p for trend < 0.001). When divided into two periods, the mean annual incidence between 2004 and 2008 was 3.8%, and 5.5% between 2009 and 2014, respectively (*p* < 0.001). With regard to PCI in AMI patients, first-generation DES was predominantly used until 2008. By contrast, the proportion of second-generation DES has surpassed that of first-generation from 2009, and it reached nearly 90% in 2014. Others included thrombus aspiration only, balloon angioplasty, bare-metal stent, and biodegradable stent.

### 3.4. Changes in Prevalence of Bleeding Risk Factors in AMI Population

In the AMI population, the changes in prevalence of individual bleeding risk factors, specified by the ARC consensus, is demonstrated in Figure 1B and Appendix A. Regardless of the period, age > 75, moderate anemia, and moderate CKD were the three most common among all criteria. Of the major criteria, severe anemia had the highest prevalence. Compared to the preceding period, statistically significant increases in prevalence were observed in criteria such as age > 75, moderate anemia, prior any stroke, anticoagulation, and the use of NSAID between 2009 and 2014. Of them, age > 75 showed the highest absolute increase in prevalence between the two periods. Although there was no statistical significance, the prevalence of malignancy was also higher in the later period compared to before.

### 3.5. Clinical Outcomes in AMI Patients Classified by the ARC-HBR Definition

Clinical outcomes in the ARC-defined HBR or non-HBR patients with AMI are summarized in Figure 2 and Table 2. The cumulative incidence of BARC 3 and 5 bleeding at 1 year was significantly higher in HBR than in non-HBR patients (9.8% vs. 2.9%, *p* < 0.001). Among the total bleeding events for 1 year, periprocedural bleeding accounted for 67.2% in the HBR and 77.2% in non-HBR patients, respectively (Appendix A). In subgroup analysis, the prevalence of HBR was higher in patients with NSTEMI compared to STEMI (37.7% vs. 28.6%, *p* < 0.001), and HBR patients in both the NSTEMI and STEMI subgroups commonly demonstrated higher 1-year cumulative BARC 3 and 5 bleeding rates (NSTEMI: 8.9% vs. 2.8%, STEMI: 10.7% vs. 3.0%, both *p* < 0.001) (Figure 2B,C). In those taking both aspirin and a potent P2Y12 inhibitor, the cumulative incidence of BARC 3 and 5 bleeding at 1 year was significantly higher in the HBR than in the non-HBR patients (10.3% vs. 3.0%, *p* < 0.001) (Figure 2D).

The cumulative incidence of all-cause mortality at 1 year was remarkably higher in the HBR than in the non-HBR patients (22.8% vs. 4.3%, *p* < 0.001). In both the NSTEMI and STEMI subgroups, HBR patients commonly had much higher all-cause mortality (NSTEMI: 19.3% vs. 3.4%, STEMI: 26.6% vs. 5.0%, both *p* < 0.001). In addition, composite of CV death, MI, or ischemic stroke was also more frequent in the HBR than in the non-HBR patients (22.6% vs. 5.8%, *p* < 0.001). Similarly, HBR patients in both NSTEMI and STEMI subgroups suffered more composite events (NSTEMI: 18.6% vs. 5.2%, STEMI: 27.0% vs. 6.3%, both *p* < 0.001). 

### 3.6. Discriminating Ability of the ARC-HBR Criteria in AMI Patients 

Receiver operating characteristic curves of the ARC-HBR criteria and PRECISE-DAPT score for BARC 3 and 5 bleeding at 1 year in the AMI population are shown in Figure 3A. Area under the curve of the ARC-HBR criteria and PRECISE-DAPT score were 0.668 and 0.675, respectively, and there was no statistical difference between them (*p =* 0.317). Sensitivity, specificity, positive predictive value and negative predictive value of the ARC-HBR criteria for BARC 3 and 5 bleeding at 1 year in AMI population were 58.1%, 69.4%, 12.9% and 95.4%, respectively. Within HBR patients, as the number of major ARC-HBR criteria satisfied rose, the cumulative incidence of BARC 3 and 5 bleeding at 1 year got higher (≥2 major with any minor: 14.8%, 1 major with any minor: 10.0%, ≥ 2 minor without major: 7.1%, and non-HBR: 2.9%) (Figure 3B). 

## 4. Discussion

Our study demonstrated that the incidence of major bleeding in the AMI population has increased during the past decade. Among the bleeding risk factors, the proportion of elderly AMI patients showed the greatest increase over time. In addition, the ARC-HBR criteria provided reliable predictions for major bleeding, mortality, and ischemic events in AMI patients in this study. The predictive ability of the criteria was also verified in patients with NSTEMI and STEMI, as well as in those treated with potent P2Y_12_ inhibitor. In terms of the discriminating ability, the ARC-HBR criteria and PRECISE-DAPT scores were comparable in AMI population.

It is a clinically important finding that the major bleeding incidence in AMI patients is rising in the contemporary PCI era. This finding contrasts with the result of a previous study using the Global Registry of Acute Coronary Events (GRACE) registry, which found a declining trend in the incidence of major bleeding in acute coronary syndrome patients between 2000 and 2007 [19]. Compared with the GRACE registry, the COREA-AMI registry in this study consists entirely of AMI and is later in time (2004–2014), better reflecting current clinical practice. Therefore, it appears that the differences in patient composition and changes in clinical practice such as higher rates of P2Y_12_ inhibitor use (94% vs. up to 65%) and PCI (100% vs. up to 45%) have influenced the results of two studies, although further investigation is needed. In addition, among the bleeding risk factors in our study, the proportion of elderly AMI patients showed statistically the greatest increase over time, based on absolute increase in value. Regarding the efficacy and safety of potent P2Y12 inhibitor in elderly patients, a systematic review and meta-analysis reported that potent P2Y12 inhibitor increased the risk of bleeding compared to clopidogrel in elderly patients with acute coronary syndrome [20]. However, in another meta-analysis, the treatment effect of potent P2Y12 inhibitor versus clopidogrel on efficacy and safety outcome was found to be consistent between elderly and non-elderly patients with acute coronary syndrome [21]. Therefore, although old age is considered one of the bleeding risk factors, further research is needed to elucidate the association between the rising trend of major bleeding in AMI patients and the increase in the proportion of elderly patients in the contemporary PCI era [8]. 

Previous validation studies for the ARC-HBR criteria used an all-comers PCI registry in which the proportion of AMI was less than 50% [9,10,11,12]. Therefore, these studies not only included a relatively small number of AMI patients, but also had no subgroup analysis on AMI, as well as NSTEMI or STEMI, which differ from each other in clinical and pathophysiological aspects. Although only one study evaluated the performance of the ARC-HBR criteria in AMI, it also provided no subgroup analysis on NSTEMI or STEMI and had major limitations such as single-center study, relatively small number of patients (*n* = 1391), and the use of bleeding endpoint definition other than BARC scale [22]. Furthermore, in the contemporary PCI era, the use of potent P2Y_12_ inhibitor is preferentially recommended in AMI patients and associated with a higher risk of bleeding compared to clopidogrel [23,24]. However, since the previous validation studies did not evaluate the impact of the use of potent P2Y_12_ inhibitor on the performance of the ARC-HBR criteria, questions have been raised as to whether the ARC-HBR criteria can provide reliable prediction for bleeding in AMI patients who are taking potent P2Y_12_ inhibitor. 

In our study using a large-scale, PCI-treated AMI population, the ARC-HBR criteria well identified HBR patients in the AMI population and satisfied the HBR definition of an incidence of major bleeding greater than 4% [8]. It is noteworthy that the 1-year cumulative incidence in HBR patients in our study (9.8%) was greater than the value of 7.9% reported by the Bern PCI registry study, which included all kinds of PCI-treated patients and adopted the same definition of bleeding as our study [12]. The difference in the incidence rate could be explained by the fact that the proportion of AMI patients in the Bern PCI registry was only 50%. In addition, by subgroup analysis, the ARC-HBR criteria was validated in patients with NSTEMI and STEMI, as well as in those treated with potent P2Y_12_ inhibitor. These findings are important in that STEMI patients are especially at higher risk of bleeding compared to other acute coronary syndrome, and that vast majority of AMI patients are treated with potent P2Y_12_ inhibitor in current clinical practice [5,6,7,23,24]. Considering the heterogeneous bleeding incidence rate and hazard ratio for each subgroup of AMI in our study, this implies that the performance of the ARC-HBR criteria varies depending on the clinical presentation.

With regard to mortality prediction, the performance of the ARC-HBR criteria was distinguished in AMI patients. One previous study using an all-comers PCI registry, in which the proportion of AMI was 14.4%, reported that the 1-year cumulative incidence of all-cause mortality in the ARC-defined HBR patients was 4.7% [9]. By contrast, our study showed that the 1-year cumulative incidence of all-cause mortality in the HBR patients was 22.8%, suggesting a superior ability of the ARC-HBR criteria for mortality prediction in patients with AMI than in those with angina. The predictive ability of the ARC-HBR criteria for ischemic events was also excellent in the AMI population.

The ARC-HBR criteria showed modest discriminating ability for major bleeding in AMI patients with low positive and high negative predictive value in this study. C-statistics value of the ARC-HBR criteria in AMI patients was comparable to that in all kinds of patients in the Bern PCI registry study [10]. In contrast to specificity, sensitivity of the ARC-HBR criteria was slightly lower in AMI patients than in all kinds of PCI patients, which might be attributed to relatively more frequent bleeding in the non-HBR patients with AMI than in those with other coronary artery disease. In addition, compared with the PRECISE-DAPT score, the ARC-HBR criteria had no statistical difference in discriminating ability, presenting similar statistical characteristics overall [18]. Based on these results, in current clinical practice, it is reasonable to use either system in AMI patients to predict bleeding risk.

This study had several limitations. First, the definition of major bleeding in this study is BARC 3 or 5 bleeding, which is different from that of the GRACE registry study [19,25]. Therefore, the discrepancy in the temporal trends of major bleeding incidence between the two studies may have been partly influenced by the difference in the definition of bleeding. Second, among the 17 original ARC-HBR criteria, 3 major and 1 minor criteria were not applied in this study. This limitation is almost inevitable in registry study, and also noted in previous validation studies of the ARC-HBR criteria [9,10,11]. Although the prevalence of missing criteria is estimated to be low, it may have slightly affected the predictive power of the criteria. Third, this study did not provide exact duration of DAPT, which is an important limitation. Considering the guideline’s recommendation to maintain DAPT for 1 year in AMI patients, discontinuation of DAPT within 1 year of PCI was presumably due to a bleeding event, and most of the patients who discontinued DAPT probably belonged to the HBR group [13,14,15,16]. Even taking these patients into account, 1-year BARC 3 or 5 bleeding incidence in the ARC-defined HBR patients was sufficiently higher than in the non-HBR to demonstrate the predictive ability of the ARC-HBR criteria. In this regard, developing more accurate algorithms for the duration of DAPT based on the ARC-HBR criteria and our study result may contribute to the reduction of bleeding in patient with AMI. Lastly, the study population consisted entirely of Asians. Therefore, the results may reflect ethnic differences of bleeding risk and incidence, and caution is needed to extrapolate the results to regions outside of Asia. 

## 5. Conclusions

During the past decade, the incidence of major bleeding in the AMI population has increased. As a risk stratification system, the ARC-HBR criteria provided reliable predictions for major bleeding, as well as mortality and ischemic events in patients with AMI and subgroups such as NSTEMI, STEMI, and in those treated with potent P2Y12 inhibitor. With respect to the discriminating ability, the ARC-HBR criteria and PRECISE-DAPT score were comparable in AMI population.

## Figures and Tables

**Figure 1 jcm-11-00988-f001:**
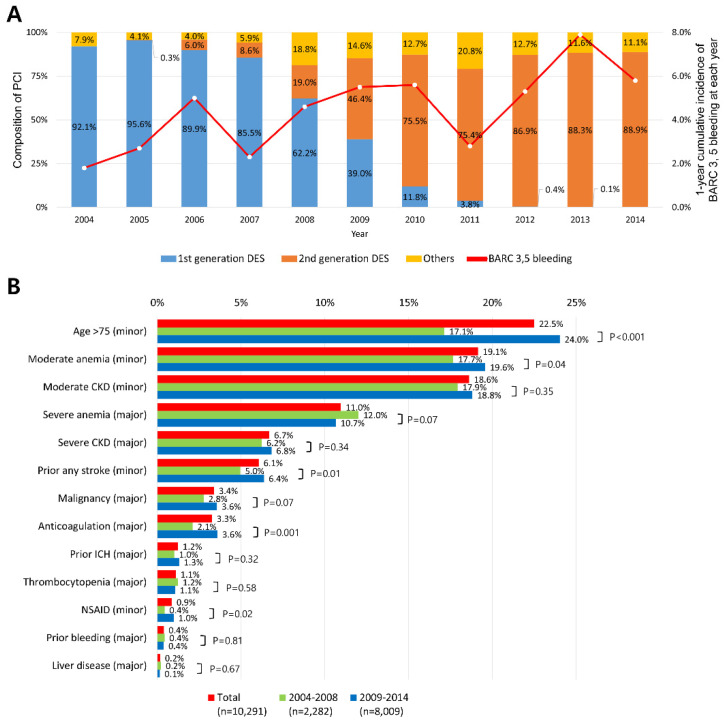
Trends in incidence of major bleeding and prevalence of bleeding risk factors in the AMI population. (**A**) The upward trend in annual incidence of BARC 3 and 5 bleeding (red line) in the AMI population is demonstrated. Composition of types of PCI in each year is expressed as bar graphs. (**B**) In the AMI population, the changes in prevalence of bleeding risk factors, specified by the ARC consensus, is shown, divided into the entire period (red bar), between 2004 and 2008 (green bar), and between 2009–2014 (blue bar). Abbreviations: AMI, acute myocardial infarction; ARC, academic research consortium; BARC, bleeding academic research consortium; CKD, chronic kidney disease; DES, drug eluting stent; HBR, high bleeding risk; ICH, intracranial hemorrhage; NSAIDs, nonsteroidal anti-inflammatory drugs; PCI, percutaneous coronary intervention.

**Figure 2 jcm-11-00988-f002:**
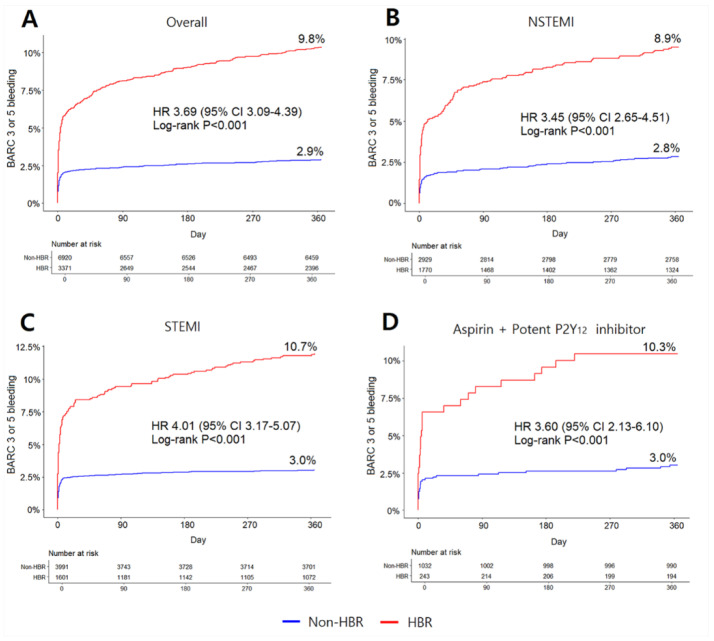
Cumulative incidence of primary bleeding outcomes at 1 year in the ARC-defined HBR vs. Non-HBR patients with AMI and subgroups. Kaplan-Meier plots for the BARC 3 and 5 bleeding incidence within 1 year after index PCI in overall AMI population (**A**), NSTEMI group (**B**), STEMI group (**C**), patients treated with potent P2Y_12_ inhibitor (**D**) are demonstrated. Abbreviations: AMI, acute myocardial infarction; ARC, academic research consortium; BARC, bleeding academic research consortium; HBR, high bleeding risk; NSTEMI, Non-ST segment elevation myocardial infarction; STEMI, ST segment elevation myocardial infarction.

**Figure 3 jcm-11-00988-f003:**
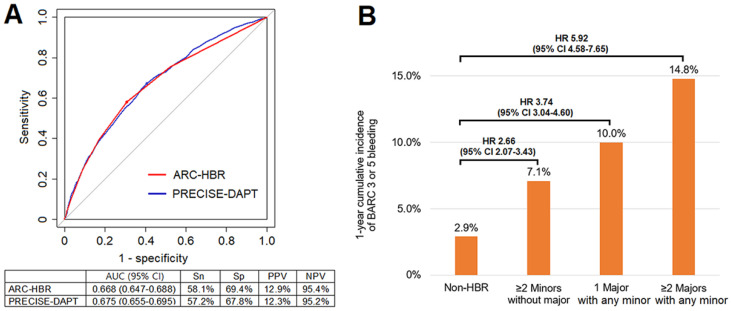
C-Statistics and additive prognostic value of the ARC-HBR criteria for prediction of BARC 3 and 5 bleeding at 1 year. (**A**) Receiver operating characteristic curves of the ARC-HBR criteria and PRECISE-DAPT score for BARC 3 and 5 bleeding at 1 year in AMI population are shown. (**B**) Within HBR group, cumulative incidence of BARC 3 and 5 bleeding at 1 year got higher as the number of fulfilled major ARC-HBR criteria increased. ARC-HBR, academic research consortium for high bleeding risk; AUC, area under the curve; BARC, Bleeding Academic Research Consortium; NPV, negative predictive value; PPV, positive predictive value; PRECISE-DAPT, predicting bleeding complications in patients undergoing stent implantation and subsequent dual antiplatelet therapy; Sn, sensitivity; Sp, specificity.

**Table 1 jcm-11-00988-t001:** Baseline characteristics.

Characteristic	HBR Group (*n* = 3371)	Non-HBR Group (*n* = 6920)	*p* Value
Demographic characteristics			
Age, years	72.7 ± 10.6	59.4 ± 11.4	<0.001
Age > 75 yrs, No. (%)	1752 (52.0)	563 (8.1)	<0.001
Female, No. (%)	1479 (43.9)	1470 (21.2)	<0.001
BMI, mean (SD), kg/m^2^	23.2 ± 3.4	24.5 ± 3.1	<0.001
Medical history, No. (%)			
Diabetes	1404 (41.6)	1849 (26.7)	<0.001
Hypertension	2298 (68.2)	3112 (45.0)	<0.001
Dyslipidemia	447 (13.3)	1188 (17.2)	<0.001
Current smoker	747 (22.2)	3347 (48.4)	<0.001
Previous MI	196 (5.8)	233 (3.4)	<0.001
Previous CABG	25 (0.7)	25 (0.4)	0.01
Atrial fibrillation	351 (10.4)	212 (3.1)	<0.001
Peripheral artery disease	36 (1.1)	23 (0.3)	<0.001
Clinical status, No. (%)			
NSTEMI	1770 (52.5)	2929 (42.3)	<0.001
STEMI	1601 (47.5)	3991 (57.7)	<0.001
Killip class			<0.001
1	1818 (59.2)	5162 (81.3)	
2	372 (12.1)	478 (7.5)	
3	413 (13.4)	236 (3.7)	
4	469 (15.3)	475 (7.5)	
Procedural characteristics			
Femoral access, No. (%)	2826 (83.8)	5528 (79.9)	<0.001
Complex PCI, No. (%)	1483 (44.0)	2870 (41.5)	0.02
Number of treated lesions, No. (%)			0.48
1	2469 (73.2)	5092 (73.6)	
2	887 (26.3)	1791 (25.9)	
3	11 (0.3)	33 (0.5)	
PCI of LMCA, No. (%)	182 (5.4)	251 (3.6)	<0.001
Types of stent, No. (%)			<0.001
BMS	241 (7.1)	201 (2.9)	
DES (1st generation)	714 (21.2)	1711 (24.7)	
DES (2nd generation)	2061 (61.1)	4460 (64.4)	
Total stent number, mean (SD)	1.5 ± 0.9	1.6 ± 0.9	0.34
Total stent length, mean (SD), mm	34.0 ± 20.7	34.2 ± 20.9	0.62
Medication at discharge, No. (%)			
Aspirin	2926 (86.8)	6668 (96.4)	<0.001
Clopidogrel	2728 (80.9)	5679 (82.1)	<0.001
Prasugrel or Ticagrelor	245 (7.3)	1050 (15.2)	<0.001
DAPT	2908 (86.3)	6656 (96.2)	<0.001
Aspirin + Clopidogrel	2665 (79.1)	5623 (81.3)	
Aspirin + Prasugrel	98 (2.9)	634 (9.2)	
Aspirin + Ticagrelor	145 (4.3)	399 (5.8)	
Anticoagulation	336 (10.0)	0	<0.001
Proton pump inhibitor	628 (18.9)	887 (12.9)	<0.001

Abbreviations: AMI, acute myocardial infarction; BMI, body mass index; BMS, bare metal stent; CABG, coronary artery bypass graft; DAPT, dual antiplatelet therapy; DES, drug-eluted stent; HBR, high bleeding risk; LMCA, left main coronary artery; MI, myocardial infarction; PCI, percutaneous coronary intervention; NSTEMI, non-ST-segment elevation myocardial infarction; SD, standard deviation; STEMI, ST-segment elevation myocardial infarction.

**Table 2 jcm-11-00988-t002:** Cumulative Incidence of Primary Bleeding and Secondary Outcomes at 1 year.

	HBR (*n* = 3371)		Non-HBR (n = 6920)		Cox Analysis Hazard Ratio (95% CI)	*p* Value
	Events/Patients at Risk	Incidence Rate (%)	Events/Patients at Risk	Incidence Rate (%)		
BARC 3, 5 bleeding						
Overall	329/3371	9.8	202/6920	2.9	3.69 (3.09–4.39)	<0.001
NSTEMI	158/1770	8.9	82/2929	2.8	3.45 (2.65–4.51)	<0.001
STEMI	171/1601	10.7	120/3991	3.0	4.01 (3.17–5.07)	<0.001
Aspirin + Potent P2Y_12_ inhibitor	25/243	10.3	31/1032	3.0	3.60 (2.13–6.10)	<0.001
Aspirin + Clopidogrel	232/2665	8.7	142/5623	2.5	3.63 (2.95–4.47)	<0.001
All-cause mortality						
Overall	768/3371	22.8	298/6920	4.3	5.84 (5.11–6.68)	<0.001
NSTEMI	342/1770	19.3	100/2929	3.4	6.17 (4.93–7.71)	<0.001
STEMI	426/1601	26.6	198/3991	5.0	6.06 (5.12–7.17)	<0.001
Aspirin + Potent P2Y_12_ inhibitor	29/243	11.9	13/1032	1.3	9.98 (5.19–19.21)	<0.001
Aspirin + Clopidogrel	346/2665	13.0	111/5623	2.0	6.99 (5.64–8.66)	<0.001
CV death/MI/Ischemic stroke						
Overall	762/3371	22.6	403/6920	5.8	4.32 (3.83–4.87)	<0.001
NSTEMI	329/1770	18.6	152/2929	5.2	3.93 (3.25–4.77)	<0.001
STEMI	433/1601	27.0	251/3991	6.3	4.90 (4.19–5.72)	<0.001
Aspirin + Potent P2Y_12_ inhibitor	29/243	11.9	33/1032	3.2	3.87 (2.35–6.38)	<0.001
Aspirin + Clopidogrel	381/2665	14.3	203/5623	3.6	4.24 (3.58–5.03)	<0.001

Abbreviations: BARC, Bleeding Academic Research Consortium; CV, cardiovascular; HBR, High Bleeding Risk; MI, myocardial infarction; NSTEMI, Non-ST segment elevation MI; STEMI, ST segment elevation MI.

## Data Availability

The datasets used and/or analyzed during the current study are available from the corresponding author on reasonable request.

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
