# Peer review of "Temporal Trends of Major Bleeding and Its Prediction by the Academic Research Consortium-High Bleeding Risk Criteria in Acute Myocardial Infarction"

_jcm, 2022, doi:10.3390/jcm11040988_

Round 1

Reviewer 1 Report

Dear Authors,

            The results presented in the manuscript "Temporal trends of major bleeding and its prediction by the Academic Research Consortium-High Bleeding Risk criteria in acute myocardial infarction" are extremely important and interesting. From the clinical point of view, a detailed knowledge about the risk of major bleeding incidents in patients after AMI undergoing PCI is very important.

            The presented paper provide evidences, that the ARC-HBR criteria provided reliable prediction for major bleeding, mortality and ischemic events in patients after AMI treated by dual anti-platelet therapy. Another important finding is that the major bleeding incidence is rising in the contemporary PCI era.

            One of the authors' conclusions is that incidence of major bleeding in post AMI population has increased with the increase in elderly patients as well as in those treated with potent P2Y12 inhibitor. There are many publications in the available literature concerning the efficacy and safety of potent P2Y12 inhibitors versus clopidogrel in elderly patients, e.g.
Am Heart J. 2018 Jan; 195:78-85. doi: 10.1016/j.ahj.2017.09.012. Epub 2017 Sep 21 and Am Heart J. 2021 Jul; 237:34-44. doi: 10.1016/j.ahj.2021.03.009. Epub 2021 Mar 15.
The manuscript should be extended  to discussion above publications.

Reviewer 2 Report

A very important study for clinical practice, but several questions arose, For example, is it known for what indications proton pump inhibitors were prescribed; it is interesting to find out the sources of bleeding that developed, which of them were more common. Of course, the main limitation of the study is the unknown exact duration of the use of double antithrombotic therapy. This study may serve to develop more accurate algorithms for the duration of double antithrombotic therapy and reduce the risk of bleeding in myocardial infarction
